# Biosecurity and Lairage Time versus Pork Meat Quality Traits in a Farm–Abattoir Continuum

**DOI:** 10.3390/ani12233382

**Published:** 2022-12-01

**Authors:** Ivan Nastasijevic, Milica Glisic, Milan Milijasevic, Sasa Jankovic, Radmila Mitrovic, Jelena Babic Milijasevic, Marija Boskovic Cabrol

**Affiliations:** 1Institute of Meat Hygiene and Technology, 11000 Belgrade, Serbia; 2Department of Food Hygiene and Technology, Faculty of Veterinary Medicine, University of Belgrade, 11000 Belgrade, Serbia

**Keywords:** pig chain, farm, biosecurity, biomarkers, stress hormones, acute phase proteins, meat quality

## Abstract

**Simple Summary:**

The modern pig meat production chain is increasingly focused on farm biosecurity, animal health, meat quality, and safety. The substantial increase of the global pork meat consumption is associated with many challenges that have impact on world meat markets, such as animal disease outbreaks, sanitary restrictions, and trade regulations and policies. To overcome such challenges and assure more consistent pork meat quality (and safety), there is a need to develop effective and reliable monitoring system in a farm–abattoir continuum. The use of specific biomarkers for proactive assessment of the pig chain characteristics is therefore of importance. Namely, the introduction of new user-friendly methodology, based on selected biomarkers, e.g., stress hormones and inflammation markers (acute phase proteins) may contribute to evaluation of the quality of processes, detection of critical points in the production system, and may anticipate the pork meat traits (quality and safety). In addition, such an approach can also contribute to the better understanding of interrelations between farm biosecurity and animal health versus meat quality and safety and can be used for detection of failures in the pig production system and with potential to be incorporated in certification programs for the pork meat industry.

**Abstract:**

The modern pig production chain is increasingly focused on biosecurity, quality, and safety of meat and is associated with many challenges impacting world meat markets, such as animal disease outbreaks and sanitary restrictions, trade regulations and quality requirements. To overcome such challenges and assure more consistent pork meat quality (and safety), there is a need to develop an effective and reliable monitoring system in a farm–abattoir continuum that can be based on selected biomarkers. This study assessed interrelations of selected stress and inflammation biomarkers (acute phase proteins (APP)) between farm biosecurity score versus pork meat quality traits after two different lairage periods. Briefly, the maximum recorded levels of stress hormones (436.2 and 241.2 ng/mL, for cortisol and Chromogranin A (CgA), respectively) and APP (389.4 and 400.9 μg/mL, Pig Major Acute Proteins (MAP) and Haptoglobin (Hp), respectively) at four commercial farms were within the recommended threshold values. Cortisol and APP were negatively correlated to the internal and total biosecurity scores of farms. The increase of level of both sets of biomarkers was found at bleeding (after transportation and lairage period), but with lower values after long (18–20 h) versus short (1–3 h) lairage lay-over time. In general, negative correlation was confirmed between stress and inflammation biomarkers and carcass/meat quality traits. The farm total biosecurity level significantly affected chilling yield, meat temperature, and a* value. Pig-MAP emerged as a good biomarker with a promising potential for assessment and anticipation of broad aspects in the pork meat chain. It can be used for detection of failures in the pig production system and might be incorporated in certification programs for the pork meat industry.

## 1. Introduction

The global consumption of pork meat has increased substantially over previous decades in spite of a temporary decline in 2019 due to the outbreak of African Swine Fever (ASF) and its global spread, mainly in wild boars, with some reported outbreaks in commercial farms, in Africa, Central Europe, Russia and Asia (China and Southeast Asia). It is projected to increase to 127 Mt over the next ten years, which accounts for 28% of the total increase in meat consumption [1]. In the EU, production increased for all categories of meat, especially pork meat, because of the robust import demand from China. The main factors that influence the evolution and dynamics in world meat markets, including pork, are animal disease outbreaks, sanitary restrictions, trade policies and quality requirements [1]. It is therefore of vital interest for the industry to apply best production practices in the farm–abattoir continuum to ensure the health status of pigs, as well as the safety and quality of pork meat, which will match the consumer preferences. Moreover, consumer perception based on cultural, ethical (e.g., animal welfare) and environmental dimensions related to pork production, including geographical origin, are important from a societal point of view [2].

The pork industry is strongly committed to assuring the integrity and consistency of fresh pork meat quality in the supply chain [3]. New questions have been raised regarding the health and welfare of animals, as well as the development of the suitable procedures and tools for their rapid and objective assessment at each stage of the food chain [4]. Significant investments and research efforts were made in previous decades to develop and optimize a system that can provide desirable and consistent quality of pork meat. It is known that fresh pork meat quality traits can be related to the animal welfare status on-farm, as well as a variety of other factors such as genetics, production system, transportation, slaughter methods, carcass dressing and post-mortem ageing [2,5]. However, there is a scarcity of data related to the impact of farm biosecurity, associated stress and general health status of pigs in the farm–abattoir continuum to the pork meat quality traits. A poor farm biosecurity level may lead to the occurrence of acute or chronic infections in fattening (post-weaning, grow-finish) pigs induced by various agents of bacterial, viral and parasitic origin causing central nervous system diseases (Glässer’s disease, edema disease, Aujezsky’s disease), gastrointestinal (salmonellosis, collibacillosis, swine dysentery, porcine epidemic diarrhea, transmissible gastroenteritis, rotavirus infection), respiratory (actinobacillosis, mycoplasmosis, porcine reproductive and respiratory syndrome (PRRS), swine influenza, atrophic rhinitis, ascaridiosis), skin (staphylococcosis, erysipelas, sarcoptosis, porcine dermatitis and nephropathy syndrome (PDNS)) and musculoskeletal diseases (osteochondrosis, toxoplasmosis) [6]. These acute or chronic conditions, including tissue injury, nutritional deficits, stress or neoplastic growth, can lead to the development of acute phase reactions by the immune system involving nonspecific early response of the organism accompanied by acute-phase proteins (APP) induction, such as Haptoglobin (Hp) and Pig Major Acute Proteins (Pig-MAP) [7,8,9,10,11]. By definition, APP are proteins whose plasma concentration changes at least 25% in response to inflammation, and they are crucial components of innate immune defense and the acute phase response, and also in chronic disease conditions [12,13]. Serum concentrations of APP are widely used as markers of inflammation and infection [14]. The liver is considered a major source of APP [15]. Over the last 30 years, extensive research has been devoted to understanding the synthesis and kinetics of acute-phase proteins, and it has been suggested that the measurement of these non-specific physiological biomarkers as well as a combined index of positive and negative APPs are an appropriate prognostic and diagnostic tool in veterinary medicine, particularly useful in monitoring the health status and welfare of pig herds on farms and at slaughterhouses [8,9,16,17,18,19].

There is a lack of data in the literature on the extent to which APP measurements reflect the long-term health and wellbeing of pigs on farms, and it is suggested that repeated measurements and combined assessment of multiple APP could provide better insight into herd health and welfare status [20]. Some studies which observed the correlation between APP and carcass or meat quality are conducted only under controlled conditions or include individual acute phase protein, but there is a need for validation of this relationship between several main inflammation markers (APP) specific for pigs (e.g., Hp, Pig-MAP) and carcass and meat quality traits. This is also implied for stress hormones (e.g., cortisol and Chromogranin A-CgA) and their interrelation with pigs’ carcass quality traits [21,22,23]. Furthermore, there are somewhat inconsistent findings in the literature regarding the influence of lairage lay-over time, as an integral part of commercial meat industry practices, on pigs’ carcass and meat quality. In some publications, shorter lairage times are recommended for pigs, e.g., between 1–3 h [24] or between 3–18 h [25], while there are also opposite findings in other studies observing that the longer lairage time (more than 18 h, but less than 22 h), although more stressful and detrimental to carcass quality, can also lead to the better meat quality traits compared to short lairage time [26,27,28,29]. Therefore, early detection of selected inflammation biomarkers—APP (Hp, Pig-MAP)—as well as stress biomarkers—hormones (Cortisol, CgA)—in commercial fattening pigs’ environment within the farm–abattoir continuum can provide valuable and early information about the farm biosecurity level, general animal health status and welfare status of pigs, and can be of great value for the pork industry, since the carcass and meat quality traits can be predicted and corrective measures can be carried out on a timely basis to prevent deterioration of meat quality. Validation of such a protocol based on the determination of selected APP and stress hormones’ levels in the blood of fattening pigs on-farm and at the abattoir (at bleeding) can improve assessment of the farm biosecurity level, general pig health status and serve for prediction of the pork carcass and meat quality traits.

To our knowledge, measurements of Hp and Pig-MAP biomarkers together with cortisol and CgA in pigs’ blood, as well as interrelation between farm biosecurity (associated with general animal health status) and meat quality traits, were not reported before in the available scientific literature in a comprehensive way. Therefore, the hypothesis applied in this study is that the farm biosecurity score has an impact on the general health status of fattening pigs in the commercial farming system, and also on meat quality traits, in the context of the farm–abattoir continuum. In addition to that, two different, short and long pre-slaughter lairage lay-over time protocols (1–3 h and 18–20 h, respectively) were assessed to reflect common practices applied in commercial operations by the meat industry, depending on the time of arrival of pigs and the time schedule of the abattoir, as no specific regulatory requirement on pre-slaughter lairage time is defined.

This study aimed to highlight the following aspects: a) determine the impact of the biosecurity score on the farm and lairage time in the abattoir on the immune response (APP concentrations: Hp, Pig-MAP) and stress (Cortisol, CgA) of fattening pigs in commercial farms’ settings and at bleeding point in the abattoir; b) validate the interrelation between farm biosecurity score, lairage time (short and long) and selected pork carcass/meat quality traits.

## 2. Materials and Methods

### 2.1. Ethics

The welfare of finishing pigs on selected farms that were used for the purposes of research was respected according to the ARRIVE checklist and recommendations regarding transparency of reporting in animal research publications [30]. In this study, no particular clinical research has been carried out, without any oral or parenteral administration of agents or any other similar procedures. The protocol applied for blood collection from fattening pigs in the commercial farm environment just followed the internal practices at four selected farms. It was carried out in parallel with the regular diagnostic procedures (serological testing intended for swine health management) applied on farms according to defined frequency, to minimize the additional disturbance and stress in pigs. Blood collection from selected pigs at the abattoir was carried out at bleeding point (after stunning), following the standard operating procedures applied at the slaughter line.

### 2.2. Farms

The experiment was conducted within a three-month period (May–July) encompassing four commercial fattening pigs’ farms, belonging to a single company that also owns an abattoir. Farms were located in northwestern Serbia, with a short distance from the farms to the abattoir, i.e., Farm 1: 13 km, Farm 2: 15 km, Farm 3 and Farm 4: 33 km. At all four farms, pigs are kept in an intensive commercial fattening system that lasts 90 days. The piglets are transferred to the fattening farm at 25–30 kg live weight, 2.5–3 months old, so that by the end of the fattening cycle, the finishing pigs achieve 115–120 kg of live weight. The housing system in the finishing farm facilities at all four farms was similar, with partially slatted floors, up to 20 pigs per pen (with a stocking density of 1 m^2^/pig). The pigs had ad libitum access to feed and water via an automatic feeder and two nipple drinkers per pen. The fattening pigs in the assessed farms were predominantly cross-breed Duroc–Landrace. The average age of examined pigs was 160–180 days. For assessment of inflammation and stress biomarkers’ variables, 100 pigs of both genders (per lairage time/per farm) were tested at four selected farms for Protocol A, which included a short retention time in abattoir lairage: 1–3 h (n = 400) and Protocol B, which considered long retention time in abattoir lairage: 18–20 h (n = 400).

### 2.3. Biosecurity and on-Farm Sampling

The biosecurity level at four commercial fattening pigs’ farms was conducted using the ‘Biocheck.Ugent’ (Biocheck Pigs, Scarborough, CU, USA) scoring checklist for measuring and quantifying the level of biosecurity in pig farms [31]. This tool is composed of relevant components of farm biosecurity and is subdivided into several subcategories, as follows: (A) Purchase of breeding pigs, piglets and semen; (B) Transport of animals, removal of carcasses and manure; (C) Feed, water and equipment supply; (D) Visitors and farm workers; (E) Vermin and bird control; (F) Location of the farm; (G) Disease management; (H) Farrowing and suckling period; (I) Nursery unit; (J) Finishing unit; (K) Measures between compartments, working lines and use of equipment; (L) Cleaning and disinfection. Using this checklist, it is possible to quantify all information on each abovementioned aspect of biosecurity.

The blood samples (10 mL per sample) for determination of APP and stress hormones’ levels were taken from randomly selected 30 fattening pigs per farm for Protocol A and B (n = 240). Blood was taken by jugular venepuncture (BD Vacutainer^®^ push button blood collection set, Mississauga, ON, Canada) in accordance with the internal practice applied for blood sampling for diagnostic purposes regarding animal health monitoring (large pigs > 15 kg) on the selected four commercial farms. The time of sampling for blood collection on the farm was about two weeks before transport, usually early in the morning. The rectal temperature of all examined pigs was also recorded (thermometer ‘Beurer FT09’, Germany).

### 2.4. Transport, Lairage, Bleeding

The behavior of fattening pigs on farm at loading (four farms, 100 pigs per farm for protocol A and B; n = 800) to the trailer (from a pre-determined start gate in the farm alley until the trailer gate) and at abattoir (unloading from the trailer to the lairage, from lairage to stunning) was assessed using the Delphi approach by three experienced observers who monitored transport conditions (loading duration, stocking density, transport duration to the abattoir) and pig behavior (percentage of animals slipping and vocalization) during loading to the trailer and unloading at abattoir [32]. The total time taken to move pigs from the start gate through the gate of the trailer and handlers’ interventions needed to move the pigs forward was also recorded, including a number of handler interventions, use of board, paddle and voice.

#### 2.4.1. Transport

The transport conditions of fattening pigs from the selected four farms to the single abattoir were similar, with 100 animals per vehicle/0.485 m^2^ per animal, while electric hound, PVC rod and PVC board were used during loading.

#### 2.4.2. Lairage

At the abattoir, pigs from each farm for both protocols (A and B) were moved from the trailer via unloading ramp using PVC rod and PVC board (with a downwards slope of 15°) to the dock from where pigs entered a 12-m long corridor before they were placed in the lairage pens. The number of pigs per pen was 33 with an available area of 0.97 m^2^ per animal. The recorded lairage time (as per commercial practice carried out by the food business operator, depending on the arrival of pigs intended to slaughter and the predicted time of the slaughter process) was 1–3 h and 18–20 h, for Protocols A and B, respectively. The pigs had free access to the water during lairage lay-over time, prior to slaughter.

#### 2.4.3. Stunning and Bleeding

Following protocols A and B, pigs were handled to move from lairage pens to the stunning box via a 20-m corridor with an upwards slope of 12°. Sticks and electric prods were used to handle pigs from lairage pens to the stunning box. A head–heart electrical stunning system (1.8 A, 100 Hz, 4.5 s) was enacted with a three-electrode system (2 electrodes placed in the head and one electrode in the heart area), to improve stunning effectiveness [33]. After the stunning, sticking and bleeding were carried out and pigs were hoisted on a rail, prior to the other standard operating procedures applied in the meat industry. Time that elapsed from stunning to sticking/bleeding varied from 1–10 s.

### 2.5. Slaughter–Chilling

At the slaughter line (speed 250 pigs/h), blood samples (n = 240) were collected at the bleeding point, after stunning/before first washing and polishing. The samples were collected (BD Vacutainer^®^, Mississauga, ON, Canada) from the bleeding wound (*truncus brachiocephalicus*) from the same pigs, per each protocol, previously identified on the farm with a black marker on the back area. Carcass scalding, evisceration, splitting, dressing and chilling were carried out according to the Good Hygiene Practices. Carcass weight (hot carcass weight) was recorded at the end of the slaughter line/before chilling, and in the cooling chamber, 24 h post-slaughter (cold carcass weight), then used to calculate the dressing and chilling yield. Muscle pH (m. *longissimus dorsi pars lumbalis*, at the head of the last rib) was measured after 1 h and 24 h post-mortem with a portable pH-meter (Testo 205^®^, Germany). The meat temperature of the carcass halves was measured 1 h and 24 h post-slaughter (Testo 905-T1^®^ penetration thermometer, Germany).

### 2.6. Water Holding Capacity (WHC)

Water Holding Capacity was determined 24 h post-chilling using m. *longissimus dorsi pars lumbalis*, sampled from thirty pork half-carcasses (50 g sample weight, at the head of the last rib, always right half-carcass) originated per each farm, for protocols A and B (n = 240), from the same pigs from which blood samples were taken at bleeding. Applied analytical steps included determination of the water content (%) and fat-free dry matter of meat, previously comminuted in a grinder (3 mm orifice plate size), in accordance with ISO 1442:2000 and ISO 1444:2000, respectively [34,35]. Parallel to that, 300 mg portion of the same meat sample was collected for determination of WHC [36]. The WHC value is expressed as the percentage of water retained in the sample (despite weight loading) to the sample water content before setting up the pressure by loading (100%). The equation used for calculation of WHC was as follows:WHC = ((m × w) − (100 × d))/(m × w) × 100
m: the mass of the sample before WHC determination (mg); w: the water content of the sample before WHC determination (%); d: the difference in the mass (mg) of the sample before and after loading (pressure); and 100: (%).

### 2.7. Color Measurements

Color of pork meat (m. *longissimus dorsi pars lumbalis*, at the head of the last rib) was evaluated using a colorimeter (Konica Minolta Chroma Meter CR-400^®^). The CIE system color profile of L*, a*, b* was measured by reflectance colorimetry using illuminant source D65, 8-mm aperture and 10° observation angle. The L* value represents lightness (L* = 0 for black, L* = 100 for white), while a* scale represents the red/green dimension, with positive values for red and negative ones for green. The b* scale represents the yellow/blue dimension, with positive values for yellow and negative ones for blue [37]. The colorimeter was calibrated throughout the experiment using a standard white ceramic tile (Y = 87.2; x = 0.3173; y = 0.3348).

### 2.8. Blood Samples Measurements

The blood samples taken by jugular venepuncture from selected fattening pigs at four farms (n = 240) and at the abattoir (n = 240) for both protocols were transported to the laboratory in a cool bin at T < 4 °C, within 2 h from sample collection. The blood samples were then subjected to centrifugation (3400 rpm/15 min) in anticoagulant-treated tubes (EDTA-treated; ThermoFisher, USA), until plasma was separated. The plasma samples were then subjected to freezing at T < −18 °C and kept for 2 months until all sessions for determination of APP and hormones were carried out.

#### 2.8.1. Acute Phase Proteins (APP)

The Pig-MAP concentration was determined by commercial sandwich enzyme-linked immunosorbent assay (ELISA), using a commercial kit (Cusabio, Houston, TX, USA) based on two anti-Pig-MAP specific monoclonal antibodies, according to the manufacturers’ instructions. This kit is based on an ELISA previously developed and validated by Piñeiro et al. [19]. Hp concentration was determined also by a sandwich ELISA kit (Cusabio, USA) as described by Sorensen et al. [38]. Pig-MAP and Hp results were expressed as microgram per milliliter of plasma (µg/mL).

#### 2.8.2. Hormones

Determination of CgA was carried out in plasma of selected pigs using commercial ELISA kit (BlueGene Biotech, Shanghai, China) based on competitive enzyme immunoassay technique utilizing an anti-CgA antibody and CgA-HRP conjugate. The testing was done according to the manufacturers’ instructions. Measurement of cortisol in pigs’ plasma was also performed by commercial ELISA kit—Cortisol Quantitative Test (Endocrine technologies, CA, USA), according to the manufacturers’ instructions. Cortisol and CgA results were expressed as nanogram per milliliter of plasma (ng/mL).

### 2.9. Statistical Analysis

For the statistics, software SPSS 20.0 (IBM, USA) was used. The results were expressed as mean ± standard deviation. The assumption of normal distribution was tested via examination of the residuals (coefficients of skewness, kurtosis, Kolmogorov–Smirnov test and Shapiro–Wilk normality test). Plasma cortisol, CgA and APP concentrations data were skewed to the right and showed normal distribution after transformation by a logarithmic function. One-way ANOVA was performed to test the effects of different farm total biosecurity score on rectal temperature, body weight and on-farm physiological biomarkers. The blood variables at bleeding point and carcass and meat quality variables were analyzed using the GLM procedure with total biosecurity score, lairage time and their interaction as fixed effects. The association between farm total biosecurity score and lairage time (independent variables) and pigs’ weight, carcass and meat quality traits (dependent variables) was assessed using univariate linear regression. All the variables with *p* < 0.25 were retained for a multivariate regression model and the interactions between independent variables were tested. The model was also assessed for multicollinearity for the multivariate analysis. The statistical differences between examined groups were determined by Tukey’s post hoc multiple comparisons test, while Student’s t-test was used to analyze significant differences between the carcass and meat quality variables due to the lairage factor. To calculate correlations, partial Pearson’s and Spearman’s correlation coefficients were applied with farm and lairage time as the control variable. Correlations were considered weak at |r| < 0.35, moderate at 0.36 ≥ |r| < 0.67 and strong at |r| ≥ 0.68 [39]. A probability value of *p* < 0.05 was considered statistically significant in all tests.

## 3. Results

The results of biosecurity status at four different farms obtained after entering the data into the online ‘Biocheck.Ugent’ questionnaire are presented in Figure 1, while scores of 12 external and internal biosecurity sub-categories are given in Appendix A.

Stress and APP variables related with pigs’ mean rectal temperature, body weight, APP and stress hormones expression levels recorded on farm are presented in Table 1.

The significantly higher mean rectal temperature was recorded in pigs originating from Farm 4 (*p* < 0.0001), where the lowest internal and total biosecurity scores were confirmed, while a significant difference in body weight was not found between farms (*p* = 0.8873). Stress biomarkers levels showed that cortisol plasma concentration was higher in the group of pigs on Farm 3 and 4 (*p* < 0.0001), with the lower biosecurity score, but a significant difference between Farm 1 (with the highest total biosecurity) and Farm 4 (with the lowest total biosecurity) was not observed (Table 1). APP biomarkers levels showed that pigs on Farm 1 had the lowest Pig-MAP and Hp concentrations (*p* < 0.0001, Table 1). At all four selected farms, the increasing values of these two variables corresponded with the lower biosecurity scores. However, the opposite finding was related to the concentration of CgA, which showed a lower dispersion of values, and significantly lower levels in pigs on Farm 2, 3 and 4, compared to Farm 1 (*p* = 0.0008).

Correlation between the biosecurity score, stress and APP concentrations in pigs at four farms is given in Table 2. The same significant, weak-to-moderate negative correlation was found between internal and total biosecurity measures and tested variables (*p* < 0.01), with the exception of CgA plasma concentration (r = 0.258, *p* < 0.01). External biosecurity was significantly positively correlated with cortisol, Pig-MAP and Hp levels (*p* < 0.01).

The transport variables (loading time, transport duration, slipping and vocalization) are presented in the Appendix A.

CgA plasma concentration at the bleeding point in abattoir, after different lairage lay-over time (short and long), was moderately to strongly correlated with all tested transport variables and lairage duration. Weak positive correlation was determined between cortisol concentration and slipping and vocalization during transport (*p* < 0.05). In addition, cortisol and Pig-MAP concentrations were moderately and strongly negatively correlated with lairage duration, respectively (r = −0.489, r = −0.785; *p* < 0.01) (Table 3).

The lower farm total biosecurity score and its interaction with lairage time on Farm 3 and Farm 4 were associated with increased cortisol plasma concentration after long lairage time and Hp concentration after both short and long lairage times (*p* < 0.05) (Table 4). On these two farms, the Pig-MAP concentrations were increased after short lairage time, but without statistical significance determined (*p* = 0.461). Regarding CgA levels, some opposite findings were observed, with a significant decrease in pigs originating from farms with lower total biosecurity scores (Farms 3 and 4) after short and long lairage lay-over time (*p* < 0.0001). Longer lairage resting was also associated with significantly lower stress hormones and APP plasma concentrations in pigs from all four examined farms (*p* < 0.0001).

The partial correlation analysis between plasma concentrations of stress and APP biomarkers demonstrated that plasma cortisol level in pigs from selected four farms, besides cortisol level at bleeding point (*p* < 0.05), was also moderately positively correlated with Pig-MAP on farm, and CgA, Pig/MAP and Hp levels at bleeding point (*p* < 0.01). Significant weak-to-moderate correlation of Pig-MAP plasma concentration on farms was observed with all the parameters measured at the bleeding point (*p* < 0.05). At the bleeding point, a week-to-moderate positive correlation was found between all stress hormones and APP plasma concentrations (*p* < 0.01) (Table 5).

The association between plasma levels of stress and inflammation biomarkers at the bleeding point associated with short and long lairage lay-over time and pigs’ carcass/meat quality traits, revealed that carcass yield and chilling yield were negatively correlated with examined plasma biomarkers (*p* < 0.05). Meat temperature at 1 h and 24 h post chilling was negatively correlated only with CgA plasma concentration (*p* < 0.01). Additionally, a negative correlation was found between meat a* and cortisol, as well as meat b* values and all examined sets of biomarkers (*p* < 0.05) (Table 6).

The variables of carcass traits and meat quality that met the assumptions and were included in the regression model and GLM analysis with farm total biosecurity score and lairage time as independent variables are given in Appendix A and Table 7. The interaction between farm total biosecurity score and lairage time was associated with significantly higher carcass weight of pigs from Farm 4 compared to the other three farms, after long lairage time (*p* = 0.004). With long lairage time, the effect on carcass yield of pigs from Farm 4 was significantly higher compared to the other three farms (*p* = 0.045). The lower total biosecurity score (Farm 3 and 4) was associated with a decrease in chilling yield for both lairage lay-over periods (*p* < 0.0001). However, the longer resting in lairage led to a better chilling yield in pigs from all four farms (*p* < 0.0001). The meat temperature 24 h post chilling was also affected by both factors (total biosecurity score and lairage time) (*p* < 0.0001 and *p* =0.036, respectively). The meat of pigs originating from Farm 2 was significantly redder after both short and long lairage periods, compared to Farm 1, 3 and 4 (*p* < 0.0001), respectively, and the lairage factor increased the meat a* value of pigs from Farm 1 (*p* = 0.006).

## 4. Discussion

In our study, assessment of biosecurity level revealed that the farms with the higher total biosecurity scores (farms 1 and 2) provided better conditions for the fattening pigs in a commercial farm environment compared to the average pig farm in Serbia (with score of 55%). All assessed farms had notably lower biosecurity scores compared to the average biosecurity score of pigs’ finishing farms in the world (71%) [31]. All four commercial farms had better scores regarding external biosecurity, while lower internal biosecurity scores were mostly associated with certain internal sub-categories, e.g., disease management; implemented measures between compartments, working lines and use of equipment; and cleaning and disinfection. Accordingly, Pandolfi et al., who collected data on 68 different pig farms, observed that the farm total biosecurity scores ranged from 40.1 to 89.5% for the pig fattening farms, with lower internal than external biosecurity scores and the low mean sub-category scores for a farrowing period, nursery and measures between compartments and use of equipment [40]. As previously explained, a better assessed external biosecurity is a consequence of a greater awareness of farmers and workers towards the risk of contamination and diseases from outside the farms, especially for the diseases regulated by control programs [41].

In the present study, cortisol, Pig-MAP and Hp levels were higher in pigs on farms with a lower biosecurity score, with determined weak-to-moderate negative correlations between these measurements, internal and total biosecurity score. Despite the significant differences observed, the increase in the values of the mentioned stress and APP biomarkers was not higher than the marginal values defined by previous studies [17,21], which are directly related to the presence of acute pathological conditions in pigs. This may indicate that pigs on farms with lower biosecurity scores were more exposed to chronic stress and inflammatory stimuli, where the host defense response is weaker with a lower increase in serum concentrations of physiological biomarkers [8]. Implications of different environmental factors and housing systems on pigs’ health are well documented [42,43]. For example, Scott et al. found a higher prevalence of lameness and bitten tails in a fully slatted system, while in a straw-bedded system, the main problems were respiratory and porcine multi-systemic wasting syndrome (PMWS) symptoms [42]. Additionally, higher levels of Hp and C-reactive protein (CRP) were reported in pigs reared in a fully slatted flooring system. In a study carried out by Pineiro et al., it was observed that the higher APP levels (including Hp and Pig-MAP) were associated with increased distress of pigs, thus having the impact on lower weight gain of pigs placed in experimental facilities [44]. Besides the procedures in pigs’ intensive production systems that are part of the regular practices and could cause psychological stress reactions, mainly including social disturbances, handling and novelty to the situation, when studying the corticosteroid responses, circadian fluctuations in basal concentration must be taken into consideration [8], meaning that cortisol blood basal concentration is generally higher in the morning than in the afternoon and evening and that exposure to stressors could affect or disrupt circadian rhythm [45]. In the studies where the stress biomarkers in pigs were investigated, it was observed that CgA showed promising potential to be used as a marker for the sympathetic–adrenal–medullary axis response (SAM) activation in different situations on farm, such as refeeding after a period of feed deprivation and after regrouping [45,46,47]. In a study carried out on single finishing pigs’ commercial farm, it was observed that the higher concentration of cortisol measured in individual pigs was not related directly to a high level of stress but rather reflected increased sensitivity of the hypothalamic–pituitary–adrenal axis to stressors [28]. However, none of these studies assessed the impact of farm biosecurity on the abovementioned biomarkers, as well as the temperature and weight gain of fattening pigs at commercial farm settings.

Further, in our study, cortisol and Pig-MAP concentrations at the bleeding point were increased after the pre-slaughter handling (loading, transport, unloading, different lairage times and stunning) in comparison to the values of these variables on farms. The increased stress biomarkers after transport and different lairage time (0–6 h) compared to these values on farms were also confirmed by the previous studies [48,49]. Rey-Salgueiro et al. indicated that cortisol and corticosterone levels increased after 4 and 6 h of lairage resting, suggesting the exposure of pigs to high stress [48]. Apart from transportation and lairage lay-over time, stunning (as a method to facilitate handling and reduce stress) is also an important pre-slaughter stress factor, especially if it is done improperly. Purnama et al. observed a significant increase of serum and salivary cortisol levels compared to before and after stunning [50]. Although the stunning method was applied carefully by trained operators in the same manner to all slaughter pigs included in this study, it should be taken into consideration that it also contributed to the general increase of the stress hormones and APP concentrations at the moment of bleeding. Moreover, it was observed that the plasma levels of both sets of biomarkers, e.g., stress hormones (cortisol, CgA) and APP (Hp, Pig-MAP), were lower at the bleeding point after a long versus short lairage lay-over time, at all four commercial farms. Namely, in commercial practice, the feed could be deprived for 1 to 24 h, depending on the in-field condition regarding pre-slaughter stress and the fasting time needed to reduce the volume of gut contents. However, the pigs become more physically active with prolonged feed deprivation, exhibiting a higher incidence of aggressive and mounting behavior [51]. Unlike the above mentioned, Brown et al. found that feed deprivation for up to one hour, 12 h or 18 h, had no effect on blood cortisol levels in pigs at slaughter [51]. In addition, it has been shown that fasting for 6 h, as well as prolonged feed deprivation for 24 to 26 h, did not negatively affect carcass traits and meat quality, but reduced the incidence of PSE meat without increasing the occurrence of DFD meat [48,52]. This significant decrease in levels of biomarkers at bleeding observed in our study after long lairage time supports earlier findings which confirmed that stress initiated by loading and unloading after short journeys and mixing of the animals in a lairage contributed to the increase of plasma levels of above-mentioned stress parameters, supporting the observation that the longer resting is needed for pigs to recover [53]. Averos et al. also confirmed that different serum parameters were the highest immediately after transport of pigs and decreased at the end of lairage [54]. In other studies, it was confirmed that the fighting between pigs due to socialization is commonly associated with increased cortisol plasma levels in combination with a short lairage lay-over time, e.g., <3–4 h [55,56]. The opposite observation was stated by Dokmanovic et al. who did not find significant differences between cortisol levels in short and long lairage times and confirmed that pigs that were kept in lairage for a long time had a higher blood lactate levels and potential deterioration impact to the meat quality, due to increased stress associated with rough handling, fights/bites, feed deprivation, etc. [28]. In general, increased cortisol concentration at the bleeding point at the abattoir, after the transport and lairage lay-over in the present study, was more pronounced in pigs originating from the farms with the lower assessed biosecurity level. This is supported by the findings of Jong et al., who confirmed that salivary cortisol level after transport and at the end of the lairage period in pigs reared in a barren environment was significantly increased compared to cortisol concentrations in the home pen, while this was not reported for enriched-reared pigs [57].

It is worth of noting that there is a wide range of Hp and Pig-MAP cut-off values for the differentiation of healthy animals from those with pathological conditions. Klauke et al. defined marginal values above 800 μg/mL and 700 μg/mL, for Hp and Pig-MAP, respectively, that indicate an increased risk for organ findings during meat inspection [21]. Chen et al. reported a Hp concentration of 1320 μg/mL in clinically normal pigs and 1430 μg/mL in pigs with gross lesions [17]. As for Pig-MAP cut-off values, experimental studies reported a range between 450 and 1500 μg/mL [58]. In the present study, Hp and Pig-MAP values in pigs on the selected four farms and at the abattoir (bleeding point) after different lairage periods were between 73.3 and 400.9 μg/mL and 108.8 and 389.4 μg/mL, respectively. Therefore, it can be concluded that in our study, in spite of different farm biosecurity levels, the selected pigs intended for slaughter sourced from four different farms were clinically healthy with a good general condition and without major injuries, lesions and stress that could cause an increase of indicators above the previously reported cut-off limits. However, although the significant increase in the levels of inflammation biomarkers was within the cut-off limits, our results indicate that the different farm biosecurity scores still influenced pork meat quality traits.

Cortisol, as a most commonly used stress marker in slaughter pigs, is well studied in many different contexts (slaughter season, circadian rhythm, sex, genetics, transport, feed and water deprivation, handling, lairage time, social isolation, crowding or mixing, stunning methods, carcass and meat quality) [54,59,60,61,62]. On the other hand, APPs and CgA concentrations as alternative stress biomarkers in the serum/plasma or saliva of pigs are barely investigated, mostly in association with the cortisol concentration [20,23,44,47,63,64]. The absence of a significant difference in cortisol concentration on farms with the highest and lowest biosecurity scores in our study is similar to the findings of Dokmanovic et al., who observed that increased cortisol level was not directly related to the higher level of stress [28]. On the other hand, the increase in cortisol level at bleeding caused by transport conditions (slipping and noise), shorter lairage period and stunning practice is confirmed by the results of this study. Dokmanovic et al. observed that the plasma cortisol level in pigs in a long lairage lay-over time (overnight) was lower in both cases regarding gentle and rough handling of pigs and confirmed that the cortisol level as a stress biomarker had lower concentrations in a longer lairage lay-over time, e.g., <24 h [28].

There is a scarcity of data when it comes to the specific relation between stress and APP indicators in the plasma of fattening pigs in a commercial farming system. In accordance with the results of our study, Chen et al. found increased Hp levels in pigs originating from farms with the lower biosecurity score, and that was associated with animals showing clinical signs and inflammation [17]. Further, in the present study, cortisol levels at the farm and at the bleeding point were weakly to moderately correlated with both Pig-MAP and Hp concentrations at bleeding. Earlier studies also found positive correlations between cortisol and Hp levels in serum and saliva of slaughter pigs under standard marketing conditions or exposed to different stress factors (mixing and feed deprivation) [23,47]. Such findings pinpoint the interrelation between APP and stress biomarkers, showing that the general health status of pigs in a farm-to-abattoir continuum (APP levels) is connected to stress on the farm, during transport and at abattoir lairage. Most conducted studies investigated Hp and CRP, and there is a lack of data related to APP more specific for pigs such as Pig-MAP [17,42], as this is one of the main APP in swine and it is a glycoprotein unique to this species [65,66]. In addition, more data are known regarding the level of stress hormones (cortisol, CgA) and their impact on meat quality; these biomarkers were assessed in relation to the farm production system and animal health, and lairage time at the abattoir, for their impact on carcass composition and pork meat quality traits [45,60,67]. This is particularly related to the correlation between Pig-MAP on farm/at bleeding and cortisol on farm from the present study. However, it is observed that a Pig-MAP can be used as a good biomarker of pig health status since it is the most sensitive, among the other acute phase proteins studied, in the general detection of disease [68]. In addition to these findings, Pineiro et al. also observed that Pig-MAP has the advantage of its low variability in a normal state compared with other APP (such as Hp) and can be effectively used for monitoring of animal health and welfare in the pig production chain [4].

Previous studies found a positive correlation between cortisol and CgA [47,69], which was confirmed by our results. CgA is considered an indicator of acute stress that is detected in blood and saliva a few minutes to a few hours after stress, while acute phase proteins are induced mostly by a nonspecific immune response and take longer to detect [47,70]. The physiological stress response expressed through cortisol, CgA and various APPs release controlled by the hypothalamic–pituitary–adrenocortical (HPA) and the SAM activity involves very complex and insufficiently investigated mechanisms [71]. Since CgA is triggered by the neuroendocrine pathway, similar to cortisol, it could be expected that CgA values will follow the trend of cortisol [69]. In our study, CgA values generally deviated from other blood parameters at each examination point. Thus, the CgA value was weakly positively correlated with biosecurity measures and moderately to strongly negatively correlated with transport variables and lairage time. However, testing of the interrelationship between biomarkers in our study revealed a significant positive correlation between CgA and Hp concentration on farm, and CgA and all examined blood variables at bleeding. Short-time changes in APP concentrations were reported by Huang et al., who found that 10 min of restraint stress significantly increased the level of Hp and Serum amyloid A (SAA) in the saliva, while after 30 min of stress, these concentrations decreased or even returned to pre-stress levels [64]. Further, Huang et al. also suggested that Hp and CgA could be better indicators for stress and physical state compared to cortisol and SAA, and in our study, a significant correlation between CgA and Hp was confirmed [64]. In addition, in our study, Pig-MAP and Hp plasma levels in pigs from all four farms increased at the bleeding point after short-distance transport and short lairage time, while after long lairage resting, a significant decrease in their concentration was confirmed. Although several studies examined Hp and Pig-MAP concentrations in pigs’ plasma, there is a scarcity of data clarifying their mutual relation in the continuum from farm to slaughter [58,72] and the results originated from this study can provide additional clarification accordingly.

A number of studies observed the interrelation between stress hormones (cortisol) and meat quality traits [28,73,74,75,76]. In our study, all investigated variables of pigs’ performance, meat quality and carcass composition were in a normal range with a weak-to-moderate negative correlation between stress biomarkers (cortisol and CgA) and the carcass and chilling yield, meat temperature, a* and b* meat values. Inflammation biomarkers’ (Pig-MAP and Hp) levels at bleeding were generally negatively correlated with the pigs’ carcass and meat quality traits, with significance in relation to chilling yield and meat b* value. In one of the few studies which linked meat quality to APP levels in pigs from 17 to 25 weeks of age, significant negative correlations were observed between Hp and Pig-MAP concentration and the carcass yield, as well as between Pig-MAP concentration at the time of slaughter and water content in m. *longissimus dorsi* [21]. Similar to our results, it appeared that Pig-MAP plasma levels were better correlated with carcass and meat quality variables than Hp levels. This finding was also confirmed by Čobanović et al., indicating the poor reliability of Hp concentration in predicting variation in pork quality traits [23].

The results from this study indicated that the lairage time and farm total biosecurity score showed an impact on carcass and meat quality traits, such as carcass weight, carcass yield, chilling yield, meat temperature and meat a* value. In another study, Dokmanovic et al. showed that the pork meat temperature (m. *longissimus dorsi pars lumbalis*) was slightly lower at longer lairage lay-over time (overnight) [28]. The results of our study indicated that a longer lairage resting (18–20 h) contributed to the more desirable meat color (prominent reddish). Although it is known that the long lairage lay-over time is usually more stressful and it may be detrimental to carcass quality, it can also lead to better meat quality traits compared to the short lairage time [28]. Sima et al. also observed that the pigs with longer lairage resting time had a more desirable, darker meat color [77]. In another study conducted by Costa et al., it was recommended that the combination of 12 h of on-farm feed withdrawal together with 6 h of lairage lay-over time seemed to be the best combination to reduce stomach content weight (feed and water) [78]. Hambrecht and Eissen stated that meat color patterns were related to the lairage resting time and they observed that the m. *serratus ventralis* was the only muscle that showed a difference regarding short (<45 min) and long lairage resting time (~3 h), with decreased redness after a short lairage resting period [79]. Perez et al. observed that a long lairage time without feed does not compromise animal welfare and meat quality [26]. In addition, in many other studies, it has been confirmed that a fasting period of less than 18 h increases the prevalence of PSE (pale, soft, exudative) meat [27,29]. On the other hand, long fasting periods (>22 h) induce muscle glycogen exhaustion and raise the risk of DFD (dark, firm, dry) meat [80,81,82].

The multivariate regression model of the results from our study showed that the total farm biosecurity score significantly influenced chilling yield, meat temperature 24 h post-mortem and a* meat value. Additionally, a significant interrelation between farm biosecurity and lairage time was recorded for carcass yield and a* meat value. To the best of our knowledge, the effect of quantitatively expressed biosecurity scores additionally related to the health and stress biomarkers on carcass/meat quality traits in the farm–abattoir continuum was not reported before in commercial settings. Having in mind the potential for the regular use of selected stress and inflammation biomarkers for the rapid and accurate assessment of farm biosecurity level (including animal health), as well as meat quality (and safety) traits, it is also of utmost importance to develop both: affordable, user-friendly point-of-care devices for quantitative detection of aforementioned parameters, which can be easily operated by farmers/abattoir staff (e.g., microfluidic biosensors), and to establish threshold values for these biomarkers, which will also become internationally harmonized to facilitate international trade [83]. Such development may lead to a more effective monitoring system and can be incorporated into the certification program for the pig meat chain.

Notably, the limitation of this study is the lack of data concerning tracking of blood stress and health parameters after exposure to each stress factor separately, e.g., loading, transport, unloading, lairage lay-over period and stunning, since only two sampling points, on-farm and at bleeding, were covered. However, it should be also taken into consideration that this study was carried out in a commercial production environment and not in a pilot facility, as with the majority of similar studies. The analyses used in the present study are valid and adequate for the encompassed sample size. However, only four commercial pigs’ finishing farms were included, so a larger sample size considering farms with a wider range of biosecurity scores and more animals (including animals with a variety of subclinical conditions) is needed to extrapolate more accurately the results obtained in the farm–abattoir continuum. In addition, the data did not allow the inclusion of all recorded carcass and meat quality traits into the regression model, but these primary results represent the first step of a grounded theory of the association between quantitatively expressed biosecurity score levels and pork quality traits that need to be completed by further studies.

## 5. Conclusions

This study revealed the complex interrelation of selected stress (cortisol, CgA) and inflammation biomarkers APP (Hp, Pig-MAP) in fattening pigs reared in a commercial production system, regarding impact of farm biosecurity to the pork meat quality traits.

Our findings, based on two points of sampling, on farm and at bleeding in the abattoir (at bleeding, all pre-slaughter factors, e.g., transport, lairage lay-over time and stunning influenced the determined levels of selected biomarkers), indicated a good potential for using selected inflammation and stress biomarkers for rapid and more accurate farm biosecurity assessment and meat quality traits anticipation. The visual-based assessment (e.g., check list) for the assessment of farm biosecurity may be well supplemented with quantitative determination of biomarkers’ plasma level. It appeared that selected APP levels are more accurate for the assessment of the total farm biosecurity versus stress biomarkers, where CgA showed positive correlation with the farm internal biosecurity score. In addition, cortisol and Pig-MAP levels proved to be better biomarkers for assessment of the pre-slaughter phase (transport, lairage lay-over time, stunning) compared to CgA. On the other hand, Cortisol and CgA plasma levels showed good correlation with meat quality traits (e.g., carcass yield, chilling yield, a* and b* value). Pig-MAP and Hp showed a negative correlation with carcass yield, chilling yield and b* value. In all, our study showed that a longer lairage resting (overnight, with lay-over time between 18–20 h contributed to the more desirable meat color, e.g., prominent reddish). With regard to APP biomarkers, Hp appeared to be useful for farm biosecurity assessment, while Pig-MAP emerged as a good biomarker with a promising potential for assessment and anticipation of broad aspects in the pork meat chain, including farm biosecurity, pig health status and meat quality traits, since it is more sensitive than Hp. Therefore, Pig-MAP can be also used for the detection of failures in the pig production system and might be incorporated into certification programs for the pork meat industry. Additionally, the development of affordable, user-friendly point-of-care devices for the rapid quantitative detection of selected stress and inflammation biomarkers in fattening pigs in a farm–abattoir continuum (e.g., microfluidic biosensors) can be the foundation of the future monitoring system used for the certified pork meat production chain. Further and deeper research is needed to clarify the complex inter-relation between stress and inflammation biomarkers versus farm biosecurity (animal health status) and meat quality and safety traits in fattening pigs reared in commercial production systems. Specific attention should be given to setting up and harmonizing threshold values to facilitate international trade.

## Figures and Tables

**Figure 1 animals-12-03382-f001:**
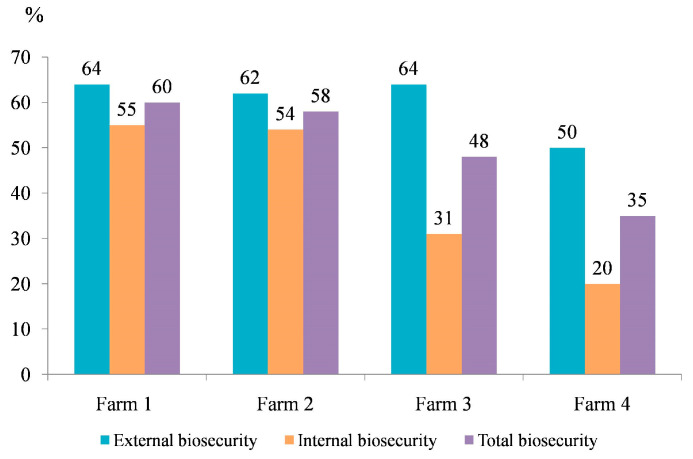
Biosecurity measures on four selected commercial fattening pigs’ farms.

**Table 1 animals-12-03382-t001:** Pigs’ mean rectal temperature, body weight and plasma concentration of cortisol, CgA and acute-phase proteins measured at four farms with different biosecurity scores.

Item	Groups				*p* Value
	Farm 1	Farm 2	Farm 3	Farm 4	
Rectal temperature (°C)	39.25 ± 0.36 ^a^	39.33 ± 0.31 ^a^	39.38 ± 0.31 ^a^	39.77 ± 0.39 ^b^	<0.0001
Body weight (kg)	119.00 ± 11.88	117.90 ± 6.49	120.20 ± 9.74	118.90 ± 9.85	0.8873
Cortisol (ng/mL) ^1^	24.17 ± 18.60 ^a,c^	21.32 ± 24.26 ^a^	81.99 ± 85.98 ^b^	60.65 ± 61.65 ^b,c^	<0.0001
CgA (ng/mL) ^1^	55.83 ± 10.11 ^a^	47.11 ± 11.21 ^b^	49.63 ± 16.66 ^b^	45.95 ± 13.94 ^b^	0.0008
Pig-MAP (μg/mL) ^1^	108.80 ± 64.52 ^a^	149.80 ± 65.22 ^b^	259.00 ± 116.20 ^c^	266.70 ± 129.40 ^c^	<0.0001
Hp (μg/mL) ^1^	138.10 ± 70.36 ^a^	173.60 ± 75.59 ^a^	358.20 ± 155.60 ^b^	318.20 ± 151.90 ^b^	<0.0001

Means ± SD in the same row followed by different lower-case superscript indicates differences ^a,b,c^ (*p* < 0.05) among farms, ^1^ Log_10_ transformed before testing.

**Table 2 animals-12-03382-t002:** Partial Spearman correlations (rs) between farms’ biosecurity measures, mean rectal temperature, stress and acute-phase proteins plasma levels (log_10_) at four farms.

	External Biosecurity	Internal Biosecurity	Total Biosecurity
Rectal temperature	−0.149	−0.434 **	−0.434 **
Cortisol	0.265 **	−0.256 **	−0.256 **
CgA	0.092	0.258 **	0.258 **
Pig-MAP	0.227 **	−0.551 **	−0.551 **
Hp	0.365 **	−0.596 **	−0.596 **

Correlations were considered weak at |rs|< 0.35. moderate at 0.36 ≥ |rs| < 0.67, and strong at |rs| ≥ 0.68; ** *p* < 0.01.

**Table 3 animals-12-03382-t003:** Partial Spearman correlations (rs) between the stress biomarkers, acute-phase proteins concentrations (log_10_ values) at bleeding point after different lairage times, transport and lairage conditions.

	Cortisol	CgA	Pig-MAP	Hp
Loading duration (min)	0.178	−0.535 **	−0.022	0.008
Transport duration (min)	0.135	−0.351 *	0.038	0.030
Slipping-transport (%)	0.207 *	−0.699 **	−0.107	−0.027
Vocalization- transport (%)	0.207 *	−0.699 **	−0.107	−0.027
Lairage Time (h)	−0.489 **	−0.748 **	−0.785 **	−0.218

Correlations were considered weak at |rs| < 0.35. moderate at 0.36 ≥ |rs|< 0.67. and strong at |rs| ≥ 0.68; * *p* < 0.05; ** *p* < 0.01.

**Table 4 animals-12-03382-t004:** Plasma concentrations of stress hormones and acute phase proteins in pigs from four farms with different biosecurity scores at the bleeding point after short and long lairage time.

Lairage Time	Groups	Variable			
		Cortisol (ng/mL) ^1^	CgA (ng/mL) ^1^	Pig-MAP (μg/mL) ^1^	Hp (μg/mL) ^1^
**Short** **(1–3 h)**					
	Farm 1	436.20 ± 143.20	177.30 ± 71.28 ^a^	292.90 ± 21.51	90.77 ± 24.38 ^a^
	Farm 2	394.80 ± 107.60	241.20 ± 61.56 ^b^	317.10 ± 83.90	188.50 ± 37.48 ^b^
	Farm 3	380.00 ± 156.80	48.17 ± 12.86 ^c^	367.10 ± 21.86	222.00 ± 104.60 ^b^
	Farm 4	323.00 ± 203.10	58.28 ± 3.44 ^c^	389.40 ± 60.82	400.90 ± 210.70 ^c^
**Long** **(18–20 h)**					
	Farm 1	169.70 ± 58.36 ^a^	44.71 ± 7.70 ^a^	143.50 ± 65.03	74.38 ± 14.11 ^a,b^
	Farm 2	121.50 ± 52.39 ^a^	57.83 ± 12.07 ^a^	201.30 ± 109.30	73.31 ± 46.47 ^a^
	Farm 3	313.10 ± 146.60 ^b^	23.21 ± 9.72 ^b^	138.60 ± 86.37	134.70 ± 80.53 ^a,b^
	Farm 4	315.70 ± 147.80 ^b^	14.43 ± 3.18 ^c^	149.70 ± 63.69	167.50 ± 108.00 ^b^
**Farm TBS**	*p* value	0.026	<0.0001	0.461	<0.0001
**Lairage**	*p* value	<0.0001	<0.0001	<0.0001	<0.0001
**Farm TBS × L**	*p* value	<0.0001	0.001	0.218	0.015

^1^ Log_10_ transformed before testing; ^a,b,c^ Different superscripts within the column indicate a significant difference between farms after the same lairage time (*p* < 0.05); TBS: total biosecurity score, L: lairage time, Farm TBS × L: interaction.

**Table 5 animals-12-03382-t005:** Partial Pearson correlation among stress and acute-phase proteins plasma levels (log_10_ values) in pigs at four farms and at the bleeding point.

	Cortisol Farm	CgA Farm	Pig-MAP Farm	Hp Farm	Cortisol BP	CgA BP	Pig-MAP BP	Hp BP
**Cortisol** **farm**	1	0.069	0.277 **	0.023	0.218 *	0.394 **	0.549 **	0.384 **
**CgA** **farm**		1	0.086	0.218 *	0.046	0.039	0.113	0.073
**Pig-MAP** **farm**			1	0.042	0.254 *	0.460 **	0.390 **	0.246 *
**Hp** **farm**				1	−0.060	0.053	0.094	−0.047
**Cortisol** **BP**					1	0.296 **	0.229 *	0.414 **
**CgA** **BP**						1	0.605 **	0.447 **
**Pig-MAP** **BP**							1	0.333 **
**Hp** **BP**								1

Correlations were considered weak at |r|< 0.35, moderate at 0.36 ≥ |r| < 0.67, and strong at |r| ≥ 0.68; * *p* < 0.05; ** *p* < 0.01; BP—bleeding point.

**Table 6 animals-12-03382-t006:** Partial Pearson correlation between plasma levels of stress hormones and APP (log_10_ values) at the bleeding point and pigs’ carcass/meat quality traits.

	Cortisol	CgA	Pig-MAP	Hp
Weight after lairage (kg)	0.216 *	0.088	0.198	0.042
Carcass weight (kg)	0.046	−0.133	−0.060	−0.118
Carcass yield (%)	−0.240 *	−0.304 **	−0.378 **	−0.226 *
Chilling yield (%)	−0.279 **	−0.457 **	−0.546 **	−0.421 **
WHC (%)	0.020	−0.054	0.005	−0.098
pH 60 min	0.099	0.004	−0.055	−0.031
pH 24 h	−0.191	−0.122	−0.059	−0.037
Temperature 60 min (°C)	0.032	−0.379 **	−0.164	−0.118
Temperature 24 h (°C)	0.001	−0.303 **	−0.155	−0.194
L* value	0.076	0.105	0.081	−0.100
a* value	−0.239 *	−0.006	−0.068	−0.133
b* value	−0.267 **	−0.425 **	−0.218 *	−0.304 **

Correlations were considered weak at |r| < 0.35, moderate at 0.36 ≥ |r| < 0.67, and strong at |r| ≥ 0.68; * *p* < 0.05; ** *p* < 0.01.

**Table 7 animals-12-03382-t007:** Effect of different total biosecurity scores and lairage time on carcass and meat quality traits.

Lairage Time	Groups	Variable				
		Carcass Weight (kg)	Carcass Yield (%)	Chilling Yield (%)	Temperature 24 h (°C)	a* Value
**Short** **(1–3 h)**						
	Farm 1	89.53 ± 13.98	76.42 ± 5.61	98.05 ± 0.12 ^aA^	5.03 ± 0.27 ^a^	6.85 ± 1.20 ^aA^
	Farm 2	85.10 ± 2.91 ^A^	75.81 ± 2.13 ^A^	98.02 ± 0.13 ^abA^	4.30 ± 0.36 ^bA^	10.11 ± 2.64 ^b^
	Farm 3	90.47 ± 3.05	75.31 ± 3.05	97.91 ± 0.10 ^bcA^	5.18 ± 0.45 ^a^	7.50 ± 1.85 ^a^
	Farm 4	87.17 ± 8.71 ^A^	75.43 ± 6.27 ^A^	97.85 ± 0.11 ^cA^	5.12 ± 0.43 ^a^	8.51 ± 1.71 ^ab^
**Long** **(18–20 h)**						
	Farm 1	87.43 ± 4.66 ^a^	79.21 ± 1.19 ^ab^	98.27 ± 0.05 ^aB^	5.05 ± 0.43	8.90 ± 1.73 ^aB^
	Farm 2	90.97 ± 2.99 ^aB^	80.36 ± 1.63 ^aB^	98.27 ± 0.07 ^aB^	4.82 ± 0.49 ^B^	12.31 ± 3.11 ^b^
	Farm 3	86.90 ± 6.11 ^a^	77.72 ± 3.48 ^b^	98.17 ± 0.12 ^abB^	5.42 ± 1.46	8.23 ± 1.09 ^a^
	Farm 4	95.97 ± 3.83 ^bB^	83.55 ± 1.15 ^cB^	98.12 ± 0.14 ^bB^	5.49 ± 0.56	8.11 ± 1.66 ^a^
**Farm TBS**	*p* value	0.269	0.045	<0.0001	<0.0001	<0.0001
**Lairage**	*p* value	0.109	<0.001	<0.0001	0.036	0.006
**Farm TBS × L**	*p* value	0.004	0.028	0.904	0.595	0.084

Means ± SD for each lairage period in the same column followed by different lower-case superscript indicates differences ^a,b,c^ (*p* < 0.05) among farms; Means ± SD for each farm in the same column followed by different upper-case superscript letters indicates differences ^A,B^ (*p* < 0.05) after short (1–3 h) and long (18–20 h) lairage resting period, TBS: total biosecurity score, L: lairage time, Farm TBS × L: interaction.

## Data Availability

Data are available upon request from the corresponding authors.

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
