# Peer review of "Biosecurity and Lairage Time versus Pork Meat Quality Traits in a Farm–Abattoir Continuum"

_animals, 2022, doi:10.3390/ani12233382_

Round 1

Reviewer 1 Report

Beside sugestions made in the pdf file, following general suggestions are given:

- title should be modified; in the manuscript is not presented just biosecurity effect, but also effect of loading and transport parameters and behaviour characteristic as well as different lairage time on stress biomarkes and acute phase protein concentrations, and pig carcass and meat quality traits

- term of biosecurity on farm refers to measures aimed to reduce probability of the introduction (external biosecurity) and further spread of pathogens within the farm (internal biosecurity) - it seems that this definition should be included in the introduction 

- in the tables should be presented real values of analsyed parameters

- discussion should be modified and should give explanations why ceratin results are obtained; disscusion should be more than presenting previous results, but givening explainations of the resutls, some statements are contradictory

- conclusion does not corespond to the results

Reviewer 2 Report

This study investigated the relationships between biosecurity and pork meat quality traits in a farm-abattoir continuum. The results are interesting and contribute to the better understanding of interrelations between farm biosecurity and animal health versus meat quality and safety. The topic is very good, but the statistical analysis needs to be improved. My comments are as follows:

Line 50, ASF from China and Southeast Asia to Africa and Central Europe? That’s not the fact. Should be from African to Europe, to Russia, to China and Southeast Asia.

Line 291, “One-way analysis of variance ANOVA was performed to test the effects of different farm total biosecurity score on physiological biomarkers and carcass and meat quality variablesOne-way ANOVA can deal with very limited information. For data analysis, I suggest that the authors should try to use the GLIMMX procedure in SAS software, which can analyze non-normally distributed data. The internal or external biosafety score, and lairage time can be used as fixed effects, and farm as random effect.

Table 1, The units of these indicators in the table were μg/mL or mg/mL. But the concentrations of cortisol, CgA and acute-phase proteins were transformed by log10. So the units in the table have to change.

In addition, this table only shows the differences among the four pig farms. I suggest that it is better to do another analysis to see the relationship between the biosecurity score and these indicators.

Table 2, Table 3, Table 5 and Table 6, the authors used Pearson's or Spearman's correlation coefficients. I suggest that the authors use Partial correlation analysis here. Because there may be complex multi-variable correlation, which needs to be corrected by partial correlation analysis.

Table 4, It is suggested to use biosecurity score and lairage time as the fixed effects and farm as random effect. And then to show these indicators between different lairage time or biosecurity score.

Table 7, please change Protocol A and B to Short or Long lairage time in the Table. It is clear that a two-factor analysis of variance was used here, but it was not mentioned in the Materials and Methods.

Round 2

Reviewer 1 Report

First of all, thanks to authors for detailed response to comments and suggestions. The manuscript has been improved and it can be accepted for publishing in present form.